# Altered immunoglobulin G glycosylation in patients with isolated hyperprolactinaemia

**Daniel Hirschberg[1], Bertil Ekman[2], Jeanette Wahlberg[2], Eva Landberg[3]***

**1** Department of Medical Biosciences, Umeå University, Umeå, Sweden, **2** Department of Endocrinology in Linköping, and Department of Health, Medicine and Caring Sciences, Linköping University, Linköping, Sweden, **3** Department of Clinical Chemistry, and Department of Biomedical and Clinical Sciences, Linköping University, Linköping, Sweden

* eva.landberg@regionostergotland.se

**Data Availability Statement:** All relevant data are within the manuscript and its Supporting Information files.

## Abstract

Prolactin is a peptide hormone produced in the anterior pituitary, which increase in several physiological and pathological situations. It is unclear if hyperprolactinaemia may affect glycosylation of immunoglobulin G (IgG). Twenty-five patients with hyperprolactinemia and 22 healthy control subjects were included in the study. The groups had similar age and gender distribution. A panel of hormonal and haematological analyses, creatinine, glucose, liver enzymes and immunoglobulins were measured by routine clinical methods. IgG was purified from serum by Protein G Sepharose. Sialic acid was released from IgG by use of neuraminidase followed by quantification on high performance anion-exchange chromatography with pulsed amperometric detection. Tryptic glycopeptides of IgG was analysed by matrix-assisted laser desorption/ionization-time of flight mass spectrometry. Hormone and immunoglobulin levels were similar in the two groups, except for IgA and prolactin. Significantly higher IgG1 and IgG2/3 galactosylation was found in the patient group with hyperprolactinaemia compared to controls. (A significant correlation between prolactin and IgG2/3 galactosylation (Rs 0.61, p<0.001) was found for samples with prolactin values below 2000 mIU/L. The relative amount of sialylated and bisecting glycans on IgG did not differ between patients and controls. The four macroprolactinaemic patients showed decreased relative amount of bisecting IgG2/3 glycans. Hyperprolactinaemia was found to be associated with increased galactosylation of IgG1 and IgG2/3. This may have impact on IgG interactions with Fc-receptors, complement and lectins, and consequently lead to an altered immune response.

## Introduction

Immunoglobulin G (IgG) is synthesized by B-lymphocytes and participates in the immune system by recognition of antigens on microorganisms, successively activating complement and binding to receptors on leukocytes to initiate phagocytosis, cytotoxic release and secretion of inflammatory mediators [1, 2]. The IgG molecule contains two light chains and two heavy chains linked by disulfide bonds. Glycans are mainly situated on the heavy chains in the CH2 domain close to the "hinge" region, which marks the passage from the variable antigen binding

**Funding:** The study was funded by the Faculty of Medicine and Health Sciences, Linköping University, Sweden.

**Competing interests:** The authors have declared that no competing interest exist.

region (Fab) to the constant part (Fc) of IgG. One N-linked oligosaccharide of biantennary complex type are usually present on Asn 297 on each heavy chain [3, 4]. In human serum from healthy individuals only a small part of N-glycans on IgG are sialylated (on average 14%), while 50% contain terminal galactose on one or two chains (IgG-G1 and–G2) and 35% lack galactose (IgG-G0) and therefore express two terminal N-acetyl glucosamine (GlcNAc) [3]. The majority of IgG Fc N-glycans are fucosylated and a small part also contain bisecting GlcNAc [4]. Both sialylation and galactosylation affect the structural stability and function of IgG. High degree of sialylation decrease the affinity for the type I Fc receptors (Fc-γRs) but increase IgG binding to the type II Fc receptors, which include C-type lectins and SIGLECs [3, 5]. Studies have shown that high amount of sialylated IgG have anti-inflammatory effects and also induces a shift of the cytokine profile [6, 7]. Agalacosyl glycoforms of IgG (IgG-G0) are increased in several autoimmune diseases, like rheumatic arthritis (RA), systemic lupus erythematosus (SLE), Crohn´s disease and Sjögren´s syndrome [3]. On the contrary, during pregnancy the agalactosylic fraction of IgG is diminished and sialylation is increased. Patients with RA often experience improvement of symptoms during pregnancy [8, 9].

Prolactin is a peptide hormone produced in the anterior pituitary and responsible for lactation and mammary gland development during pregnancy. Besides the effect of increased synthesis of α-lactalbumin, prolactin also has been found to increase the expression of β-galactosyltransferase in mammary tissue [10, 11]. Thus, prolactin may be responsible for the increased galactosylation of IgG during pregnancy. Besides during pregnancy, hyperprolactinaemia may develop due to stress, certain medications, hypothyreosis and prolactinoma, which is the most common type of hormone secreting pituitary adenoma [12]. Hyperprolactinaemia is also a common finding in patients with macroprolactin as the dominant form of prolactin. Macroprolactin is a complex between immunoglobulins and prolactin [13]. To investigate the effect of prolactin on IgG glycosylation *in vivo*, blood samples from patients with isolated hyperprolactinaemia were collected together with samples from normoprolactinaemic controls. IgG glycosylation was examined by release of terminal sialic acid by use of neuraminidase followed by quantification by high-performance anion-exchange chromatography with pulsed amperometric detection (HPAEC-PAD) and by studying the IgG-glycopeptide profiles using matrix-assisted laser desorption/ionization-time of flight mass spectrometry (MALDI-TOF MS).

## Materials and methods

### Patient selection and sample preparation

Patients with hyperprolactinaemia referred to the Department of Endocrinology at Linköping University Hospital from May 2007 to May 2013 were consecutively recruited to the study. Patients were habitants in Östergötland, a province in south east of Sweden including three cities and a surrounding countryside. The selected patients are probably representative for the Swedish population. Inclusion criteria were elevated prolactin levels found on at least two occasions (women > 465 mIU/L, men > 405 mIU/L). Exclusion criteria were pregnancy, acromegaly or untreated hypothyroidism, chronic drug abuse or doping and chronic severe disease with risk of deterioration. Finally, 25 patients with hyperprolactinaemia (22 women and 3 men) were included. Of those, 4 patients were diagnosed with macroprolactinaemia. Of the rest 21 included patients with hyperprolactinaemia, all but 2 patients underwent magnetic resonance imaging (MRI) of the sella region. Five patients had a visible microprolactinoma, 2 patients had macroprolactinoma, 2 patients had a tumour rest after previous macroprolactinoma, 1 patient had pituitary hyperplasia and 1 patient had an empty sella. The remaining 8 patients had normal MRI. The two patients not performing MRI during the study period had prolactin levels of 1090 and 2300 mIU/L, respectively.

**Table 1. Comparison of age, gender and laboratory parameters between patients and healthy controls.**

| Parameter | Patients (n = 21) | Controls (n = 22) | p-value |
|---|---|---|---|
| Mean age (±SD) | 39.2 (11.2) | 45.3 (13.9) | 0.12 |
| Women (%) | 19 (90) | 20 (91) | 0.95 |
| Haemoglobin, mean (±SD) g/L | 132 (8.1) | 135 (8.8) | 0.24 |
| Platelet count, mean (±SD) x$10^9$/L | 282 (56) | 251 (64) | 0.10 |
| Leucocyte count, mean (±SD) x$10^9$/L | 6.8 (1.7) | 5.9 (1.3) | 0.058 |
| Creatinine, mean (±SD) μmol/L | 73 (11) | 75 (11) | 0.55 |
| ALT, median (range) μkat/L | 0.33 (0.19–1.6) | 0.24 (0.10–0.53) | 0.0097 |
| ALP, median (±SD) μkat/L | 0.99 (0.37) | 0.86 (0.26) | 0.18 |
| Fasting Glucose, mean (±SD) mmol/L | 5.4 (0.49) | 5.6 (0.64)[a] | 0.25 |
| TSH, median (range) mIU/L | 2.7 (2.0) | 1.9 (1.0) | 0.10 |
| FSH, median (range) IU/L | 5.2 (0.1–78) | 9.8 (0.2–100) | 0.16 |
| Oestradiol, mean (±SD) pmol/L | 200 (160) | 220 (240) | 0.73 |
| Prolactin, median (range) mIU/L | 1090 (630–15900) | 230 (110–390) | <0.0001 |
| IgG, mean (±SD) g/L | 11.7 (1.9) | 11.6 (1.9) | 0.77 |
| IgA, mean (±SD) g/L | 2.9 (1.0) | 1.9 (0.9) | 0.0023 |
| IgM, mean (±SD) g/L | 1.5 (0.78) | 1.1 (0.44) | 0.051 |
| IgG1, mean (±SD) g/L | 5.5 (1.5) | 6.0 (1.6) | 0.34 |
| IgG2, mean (±SD) g/L | 4.7 (1.1) | 4.3 (1.1) | 0.24 |
| IgG3, mean (±SD) g/L | 0.74 (0.28) | 0.77 (0.44) | 0.77 |
| IgG4, mean (±SD) g/L | 0.38 (0.35) | 0.51 (0.45) | 0.28 |

[a]n = 18, values from four controls are missing.

A group of subjective healthy volunteers was also recruited from the province of Östergötland, Sweden between Jan 2011 and May 2016. Exclusion criteria were pregnancy and any abnormal laboratory test result according to Table 1. Finally, 22 healthy controls were included (20 women and 2 men). Blood samples were collected in the morning after one nights fasting and then again 2–3 hours later (non-fasting). Prolactin and immunoglobulins were analysed in samples from the later sampling time. Venous blood samples were collected in vacuum tubes containing coagulation activator and gel. The samples were allowed to rest for at least 30 min to complete the coagulations process and were then centrifuged at 1800 g for 10 min. Aliquots of serum were immediately stored at -20˚C until the analyses were performed.

## Prolactin analysis

Serum prolactin was measured by immunochemical detection on Cobas e602 (Roche Diagnostics, Bromma, Sweden). The calibrator is traceable to the 3rd IRP WHO reference standard 84/500. Normal range S-prolactin; women 65–465 mIU/L, men 65–405 mIU/L. Samples with S-prolactin > 400 mIU/L were checked for macroprolactin by precipitation with 25% polyethylene glycol (PEG) v:v 1:1. The samples were centrifuged at 1800 g for 10 min. Prolactin was analysed in the supernatant and the value was multiplied by 2 to correct for dilution. A recovery after PEG precipitation of less than 50% was considered positive for macroprolactin.

## Immunoglobulin analysis

Concentration of IgG subclasses and the total concentration of IgG, IgA and IgM was measured by an immune based nephelometric method on BN ProSpec (Siemens Healthcare

Diagnostics, Stockholm, Sweden). Normal range for different IgG-classes in serum are as follows; IgG 6.7–15.0 g/L, IgG1 2.8–8.0 g/L, IgG2 1.2–5.7 g/L, IgG3 0.2–1.2, IgG4 0.05–1.2 g/L.

## Other laboratory analyses

Haematological analyses, creatinine, alanine amino transferase (ALT), alkaline phosphatase (ALP), glucose, thyroid stimulating hormone (TSH), follicle stimulating hormone (FSH) and oestradiol were measured in blood, plasma or serum by routine clinical methods at the Clinical Chemistry laboratory at Linköping University Hospital.

## Immunoglobulin enrichment

IgG was enriched using a 1.5 mL Protein G Sepharose column (GE Healthcare Bio-Sciences AB, Uppsala, Sweden). The IgG enrichment from serum was carried out at room temperature (RT) by gravitational flow. The column was equilibrated with 3 mL 0.02 M PBS/0.15 M NaCl pH 7.2 (buffer A). Prior to enrichment, 200 μL serum was diluted in 1.8 mL buffer A. After application of the diluted serum the column was washed with 10 mL buffer A. Bound IgG was eluted with 3.5 mL 0.1 M glycin-HCl pH 1.8. The pH in the eluate was adjusted to 7.6 by adding 0.5 mL 1 M Tris-buffer (pH 9.5) and the column was regenerated with 5 mL 0.1 M glycin-HCl + 5 mL 0.02 M phosphate buffered saline (PBS). The column was stored at 4–8 °C until further usage. IgG concentration and purity of the eluate was estimated by bicinchoninic acid assay (BCA; Pierce, Thermo Scientific, Rockford, IL, USA) and SDS-PAGE, respectively.

## MALDI mass spectrometry

Thirty five μg IgG was diluted with 0.1 M Tris/glycin/HCl-buffert, pH 7.6 to a final volume of 100 μL. Trypsin (Sequencing Grade Modified, Promega, Madison, WI, USA), 100 μL 20 μg/mL, was added to the antibody-solution and incubated for 24 h at 37 °C. The digestion was quenched with 2 μL concentrated trifluoroacetic acid (TFA). The tryptic peptides were further desalted by solid phase extraction (SPE) using μ-C18 ZipTips (10 μL, Pierce, Thermo Scientific, Rockford, IL, USA) as follows. The ZipTip was washed and equilibrated by rinsing the SPE-column twice with 20 μL 70% Acetonitril (ACN)/0,1% TFA, twice with 20 μL 50% ACN/0,1% TFA and twice with 20 μL 0,1% TFA. Sample was applied to the column by pipetting 20 μL of the digested and acidified IgG-solution. The tryptic peptides were washed 3 times with 20 μL 0.1% TFA. To enrich glycopeptides the elution was carried out stepwise with 18% ACN/0,1% TFA, 30% ACN/0,1% TFA and lastly 60% ACN/0,1% TFA. The glycopeptides were most abundant in the first fraction and this fraction was therefore used for further analysis. The other fractions were checked for glycopeptide content. The eluates were dried down in Speedvac (Savant, Thermo Scientific) at RT for 8 min to a final volume of 1–2 μL to which 1 μL 50% ACN/0,1% TFA was added. Matrix was prepared as a 1mg/mL solution of 4-chloro-α-cyanocinnamic acid (Sigma-Aldrich) in 70% ACN [14]. Sample and matrix was mixed 1:1 and dried *in situ* on an AnchorChip-MALDI MS-plate (Bruker Daltonics) and analysed on a MALDI TOF MS (UltrafleXtreme MALDI system, Bruker Daltonics) in the positive reflector-mode. The MS was calibrated with Peptide Calibrator Standard II (REF822570; Bruker Daltonics GmbH, Bremer, Germany).

## Quantitative analysis of IgG sialylation

Purified IgG (200 μg) was added to a 10 kDa Amicon Ultra-4 centrifugal filter unit (Merck Millipore, Cork, Ireland). The filter was then rinsed two times with 2 mL 0.05 M acetate buffer (pH 5.5) by centrifugation at 4000 g for 60 min and the filtrate was discarded. Finally 200 μL

0.05 M acetate buffer was added to the top of the filter and 1 μL of neuraminidase (5 mIU; Arthrobacter Ureafaciens, Merck) was added. The solution was mixed and incubated for 48 h at 37 $^{\circ}$C. The filter unit was weighed before adding sample (empty) and after the final centrifugation, to calculate the reaction volume using 1.0 g/mL as density. The filtrate was subsequently analysed by HPAEC-PAD (ICS-3000 Dionex, Sunnyvale, CA, USA). Detectors used were an electrochemical Au-detector and an Ag/AgCl reference electrode. Twenty μL sample was injected onto a CarboPac PA-200 column (3$^*$50 mm guard column with a 3$^*$250 mm analytical column). The glycans were separated at a flow rate of 0.5 mL/min and a column temperature of 30 $^{\circ}$C, using the following setup; 0–5 min isocratic flow of 20 mM NaOH, 5–30 min linear gradient between 0 and 25 mM sodium acetate with an unchanged concentration of 20 mM NaOH. Using a 5-point calibration curve the sialic acid concentration was calculated from the area under the eluted peak in the chromatograms. An internal control material consisting of purified IgG from a normal serum pool was included in all analytical sequences. Precision, calculated as coefficient of variation, was 8.9% (n = 12).

## Ethics

The study was approved by the local Ethics Committee in Linköping (Dnr: M99-05 and 2014/404-32) and performed in accordance with the Declaration of Helsinki. The patients were informed about the purpose of the study and gave their written informed consent.

## Statistics and calculations

To detect a difference between controls and patients for the IgG-glycosylation parameters at a minimum of 20%, a relative standard deviation of 20%, alpha <0.05 and a power of 0.8, sample size was determined to 17 individuals per group. The level of galactosylation was calculated by the following equation:

$$(G1 + 2 \times G2)/(2 \times G0 + 2 \times G1 + 2 \times G2) \tag{1}$$

G0, G1 and G2 denote the fucosylated biantennary N-linked glycans with no, one or two terminal galactoses. Thus the equation represents the quote of galactose present on all available N-glycan antennae. Non-fucosylated biantennary glycans, present in minor amounts, were not included in this equation. The level of bisecting glycans was calculated by following equation:

$$(BG0 + BG1)/(BG0 + BG1 + G0 + G1) \tag{2}$$

B stands for "bisecting". Bisecting G2 glycans were not included in the calculation, due to its presence in relatively small amount or not detected in several samples. All data was checked for distribution fitting. Data with normal distributions are presented as mean +/- standard deviation, whereas data with non-normal distribution are presented as median and range. Groups were compared by two-tailed unpaired Student´s T-test (for data with normal distribution), Mann-Whitney U-test (for non-normal distributed data) and Chi-square test for categorical parameters. Differences were considered statistical significant when the p-value was less than 0.05. When applicable Bonferroni correction was used for multiple testing. P-values in Table 1 were not corrected for multiple testing as this part of the study was designed to minimize type II errors (false negatives). Correlation was examined by Pearson´s correlation test for normal distributed data and Spearman´s rank-order correlation test for non-normal distributed data. Multiple regression analysis was used to compare several predictors. The software used for statistic calculations and box plots was Statistica 13.

## Results

Samples collected from 25 patients with hyperprolactinaemia and 22 healthy controls were examined for a panel of biochemical markers. Samples from four patients with hyperprolactinaemina (520–860 mIU/L) were found to contain macroprolactin as the dominant form and corrected monomericic prolactin values after PEG-precipitation were within the normal reference interval and ranged between 160 and 260 mIU/L. These patients (three women and one man, 23–38 years) were excluded from the hyperprolactinaemia patient group. Descriptive and laboratory data are presented for each group in Table 1.

Age and gender were similarly distributed in both groups with no significant difference. The total concentration of IgA was significant higher in the patient group (2.9 compared to 1.8 g/L, p = 0.0024 or 0.043 corrected for multiple testing). Except for prolactin and ALT, no other of the measured parameters differed significantly between the groups. Glycosylation patterns of IgG was analysed by MALDI-TOF mass spectrometry of tryptic glycopeptides (Fig 1).

By this method, glycopeptides from the hinge-region of IgG1, IgG2 and IgG3 could be detected and the degree of galactosylation determined by comparing the relative intensity of the G0, G1 and G2 N-glycans. The amino acid sequence of tryptic glycopeptides from IgG2 and IgG3 is identical. Thus, the glycopeptides corresponding to m/z 2602, 2746 and 2926 represent both IgG2 and IgG3 glycopeptides (Fig 1). In IgG the majority of these N-glycans contain one fucose unit. However, non-fucosylated N-glycans also exists and peaks with m/z values corresponding to these glycans were found in minor amounts. Unfortunately, it is impossible to discriminate these from IgG4 N-glycans as they have similar m/z values. Peaks unique for IgG4 were not detected. Peaks with m/z values corresponding to bisecting N-glycan peptides from IgG2/3 were also found (Fig 1).

When using MALDI-TOF MS, the glycopeptides were analysed in the positive reflective mode, which did not allow us to simultaneously analyse sialylated glycopeptides and compare

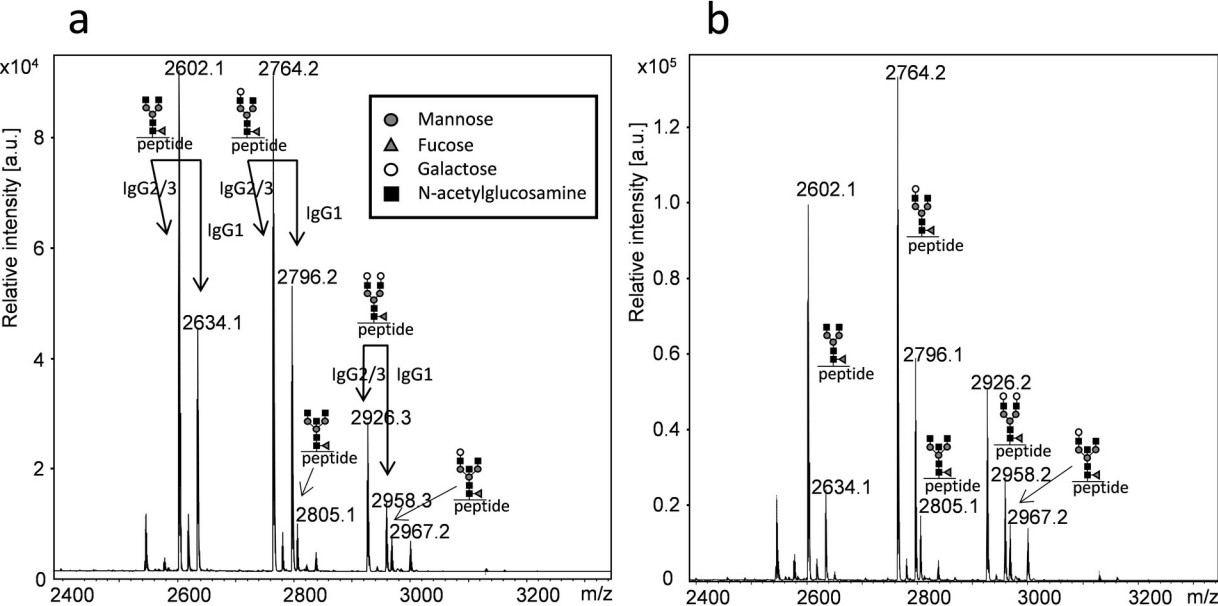

**Fig 1. MALDI-TOF MS spectra of tryptic glycopeptides from the IgG Fc part.** Analysis was performed in the positive reflector-mode using 4-chloro-α-cyanocinnamic acid as matrix. Peptides from IgG2 and IgG3 have identical amino acid sequence, thus assigned IgG2/3. Samples were collected from a normoprolactinaemic healthy individual (a) and a patient with hyperprolactinaemia (b). Symbols denote individual monosaccharides as explained in the figure.

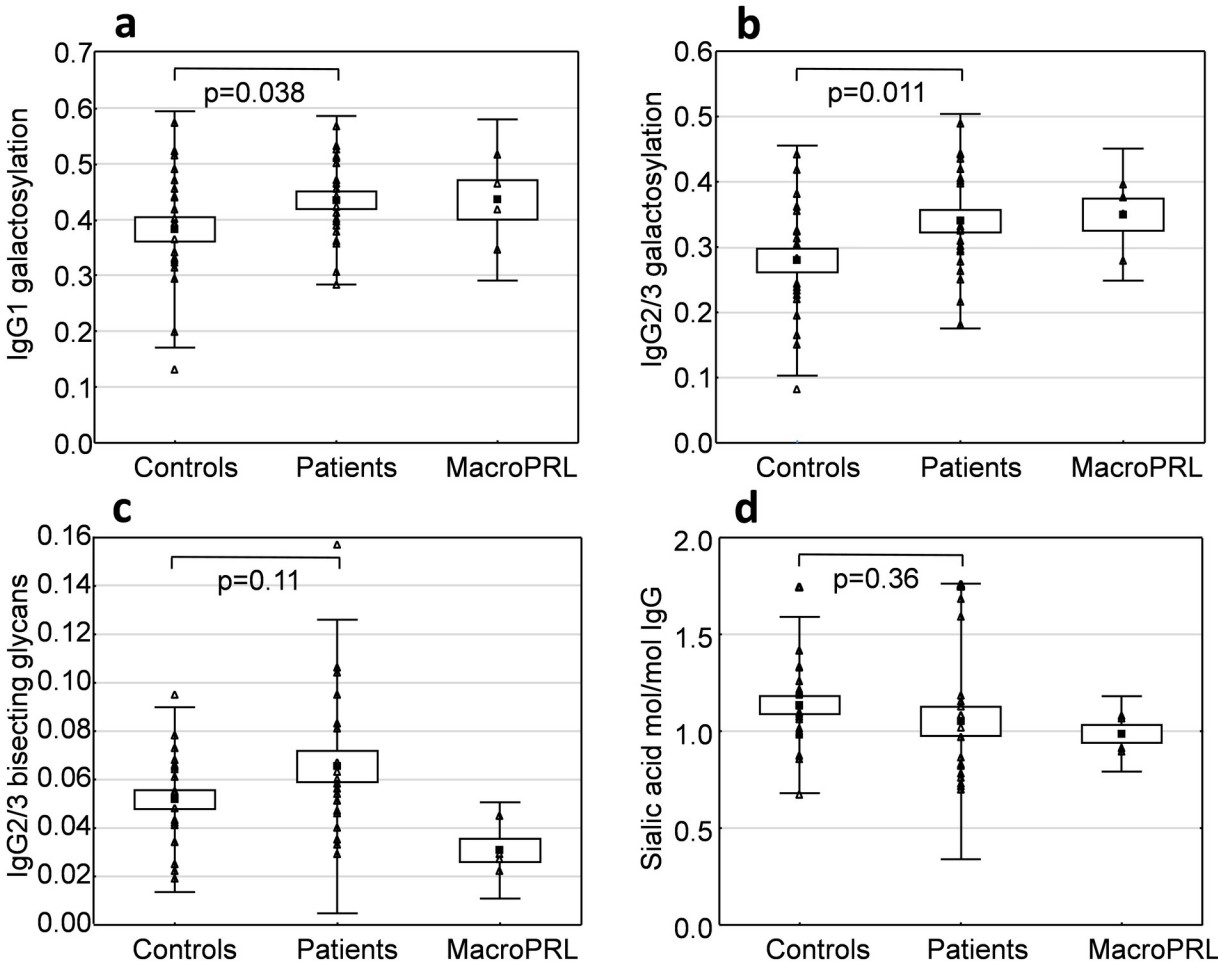

**Fig 2. Glycosylation patterns of IgG in patients with hyperprolactinaemia (n = 21) compared to healthy normoprolactinaemic controls (n = 22).** The point label indicates the mean. The box indicates standard error of mean and the bars indicate +/- 2 standard deviations. Level of galactosylation was calculated from the relative intensity of the G0, G1 and G2 N-glycans of IgG1 (a) and IgG2/3 (b). The relative intensity of bisecting glycans was calculated for the G0 and G1 N-glycans of IgG2/3 (c). Total IgG sialic acid content was measured by HPAEC-PAD after release by neuraminidase (d).

intensities [15]. In an effort to compare total IgG sialylation between groups, we used neuraminidase to cleave sialic acid from IgG glycans and further quantified the amount of released sialic acid by HPAEC-PAD (S1 Fig).

IgG galactosylation, the relative abundance of bisecting G0 and G1 glycans and IgG-bound sialic acid were calculated for controls, patients with hyperprolactinaemia and patients with macroprolactinaemia and compared (Fig 2).

Hyperprolactinaemia patients showed higher IgG1 and IgG2/3 galactosylation than the control group which was statistically significant for IgG2/3 (p = 0.011) and IgG1 (p = 0.038). There were no significant differences in the abundance of IgG2/3 bisecting glycans between the groups. Measurements of IgG sialylation indicated a high individual variation, but no significant differences were found between hyperprolactinaemia patients and healthy controls (Fig 2). The macroprolactinaemic group had lower levels of IgG2/3 bisecting glycans than patients and controls (Fig 2).

Correlation between prolactin and IgG galactosylation was further analysed.

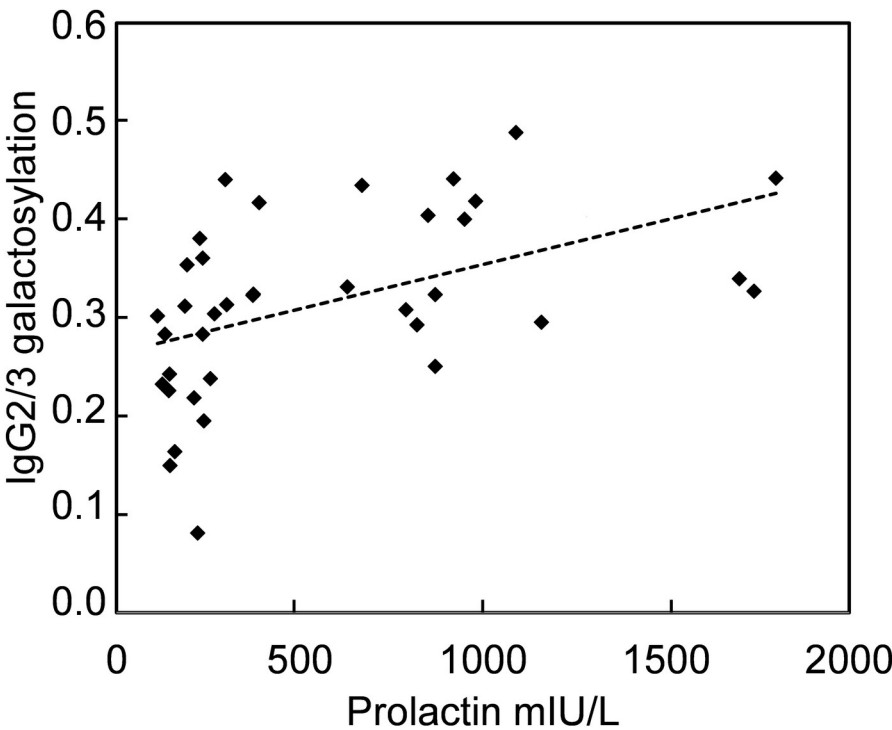

**Fig 3. Correlation between prolactin and IgG2/3 galactosylation.** Correlation between prolactin and IgG2/3 galactosylation measured in samples from controls and patients with prolactin <2000 mIU/L (Rs = 0.61, p<0.001). Samples from macroprolatinaemic patients were not included.

Even though prolactin in all samples (without macroprolactin) did correlate significantly to IgG2/3 galactosylation (Rs = 0.32, p = 0.035), excluding samples with prolactin values above 2000 mIU/L led to a higher degree of correlation with a Rs of 0.61 (p<0.001, Fig 3).

For IgG1 galactosylation, corresponding correlation indices for samples with prolactin < 2000 mIU/L were Rs = 0.29 and p = 0.081. As IgG2/3 galactosylation also was correlated to age (r = -0.59, p<0.001), multiple regression analysis was performed to compare the influence of each predictor. Prolactin values <2000 mIU/L were log transformed to approach a normal distribution (n = 37). Both prolactin and age were significant predictors of IgG2/3 galactosylation with β = 0.33 (p = 0.037) and β = -0.42 (p = 0.0095), respectively. Adjusted $r^2$ for this model was 0.40. Samples with macroprolactin were excluded from the correlation analysis because of the difficulty to determine the relevant prolactin value to use (total prolactin or corrected prolactin after PEG precipitation).

## Discussion

Hyperprolactinaemia has been associated with different autoimmune diseases, especially with SLE and also RA, multiple sclerosis, autoimmune thyreoditis, myasthenia gravis and diabetes mellitus. However in several cases there have been discordant findings [16]. At the same time altered glycosylation of the IgG Fc part is present in many autoimmune diseases, especially during deterioration [1, 3]. In RA, a decrease in IgG galactosylation is accompanying a worsening of symptoms [8, 17], whereas improvement in RA is seen for most patients during pregnancy when galactosylation as well as prolactin increases [9]. Decreased galactosylation of IgG is also found in SLE [18]. Although many studies have focused on effects of prolactin in the

immune system, until now, there have been no reports of associations between prolactin and IgG glycosylation.

This study shows that hyperprolactinaemia is associated with increased galactosylation of IgG Fc-glycans. A higher degree of galactosylation may have large impact on the immune system, as fully galactosylated N-glycans on IgG favour an open and stabile conformation of the IgG molecule with impact on bindning to Fc gamma receptors and C1q [7, 19]. On the other hand IgG with Fc glycans lacking galactose, i.e. G0 glycans, are more likely to interact with mannose binding lectins, thus activating the lectin pathway [1, 6]. Galactosylated N-glycans is also the prerequisite for increased sialylation of the IgG Fc part and increased sialylation is linked to anti-inflammatory response and development of tolerance [5, 6, 20].

IgG Fc glycosylation has also been found to depend on age. With higher age galactosylation and sialylation decrease and the amount of bisecting glycans increase [21, 22]. The change is especially pronounced in women [22]. In a recent study, the abundance of G0 glycans was examined in serum or purified IgG in two large healthy populations. The study revealed a higher G0-fraction in men and an increase of the relative abundance of G0 glycans with age, especially prominent in women after menopause. As conjugated oestrogens and raloxifene were able to reduce the amount of G0 glycans in postmenopausal women, the authors suggested a regulatory role of oestrogens in IgG galactosylation [23]. However, the importance of prolactin was not addressed in this study. Since oestrogens are known to stimulate prolactin secretion from the pituitary, there is a large covariance between oestrogens and prolactin [24, 25]. Prolactin levels are thus increased in premenopausal women but diminish to similar levels as for men after menopause. As patients and controls in our study had similar distribution of age and oestradiol levels, this indicates a causal relationship between prolactin and IgG-galactosylation, but does not exclude that oestrogens also may have these effects. We also found a significant correlation between prolactin and IgG2/3 galactosylation, but the interpretation of this finding is problematic as both prolactin and IgG galactosylation are correlated to age. Biological variation of prolactin may further obscure the connection. Prolactin secretion follows a circadian rhythm, with increased levels during sleep and early in the morning [24]. It is therefore important to collect samples at similar time of the day and at least 2–3 hours after awakening, which agree with the protocol in this study. However, prolactin shows no clinical relevant change during the menstrual cycle [26]. According to FSH levels a higher number of women in the control group had reached menopause, but since prolactinomas may depress the remaining pituitary hormone secretion and induce temporary anovulation and menstruation disruption, the group of patients may be more similar to older women regarding levels of sex hormones.

The strength of this study is that both patients and controls were examined for a large number of possible confounding biochemical factors, as markers of kidney function, haematology, thyroid status and oestradiol levels. Importantly, there were no differences in IgG levels or IgG-subclass levels between the groups.

An incidental finding was that IgA was significantly higher in the hyperprolactinaemic group compared with controls. This is in accordance with previous findings of prolactin as a stimulatory agent of IgA synthesis [27].

Increased galactosylation of the IgG Fc part is often accompanied by increased sialylation [9, 18, 20]. However, such an association was not found in this study and sialylation did not correlate to prolactin levels. A limitation is that we measured the total amount of sialic acid on IgG, thus also including sialic acid on N-glycans attached to the Fab unit of IgG. It has been estimated that Fab glycans amount to 15–20% of the IgG glycans [1]. These glycans are fully galactosylated and mostly sialylated [28]. Thus, it cannot be excluded that variation in sialylation of Fab glycans may have masked alteration of Fc sialylation in the hyperprolactinaemia

patients. In addition, sialylated O-linked glycans have been found on human IgG3 [29], which also may contribute to an increased variability of sialic acid. Prolactin may also have variable effects on sialylation on different IgG subclasses. Furthermore, glycosylation of other immunoglobulin classes in relation to hyperprolactinaemia has not been examined previously nor in this study. It would therefore be interesting to evaluate if hyperprolactinaemia is associated to similar glycosylation changes for IgA, IgM or other immunoglobulin classes.

Macroprolactin is a complex of immunoglobulins (typically IgG4) and prolactin. It is a common finding in blood, especially together with hyperprolactinaemia [13, 30]. Macroprolactin has no major clinical implications [31, 32]. The four patients with macroprolactin as the cause of hyperprolactinaemia showed an altered IgG2/3 glycosylation compared to controls and the other patients by having reduced abundance of bisecting glycans. The finding is interesting, but needs to be confirmed in a larger population. Decreased amount of bisecting glycans on the IgG Fc part has been linked to immunization [20], whereas an increase in bisecting IgG Fc-glycans is associated with increased antibody dependent cellular cytotoxicity [33]. It would also be interesting to specifically examine glycosylation of IgG involved in the binding to macroprolacin in these patients as glycomic changes of IgG may be an underlying cause of the formation of macroprolactin.

## Conclusions

Moderate hyperprolactinaemia was found to be associated with increased galactosylation of IgG1and IgG2/3. This may have impact on IgG interactions with Fc receptors, complement and lectins, and consequently lead to an altered immune response. Eventually, this knowledge may lead to new treatments options for autoimmune diseases. Further studies are needed to explore the causal relation between increased prolactin levels and IgG galactosylation. This study also showed that patients with hyperprolactinaemia have similar amounts of sialylated and bisecting glycans on IgG compared to normoprolactinaemic healthy controls. However, this has to be confirmed in a larger study discriminating between N-glycosylation of the Fc versus Fab part of individual IgG subclasses.

## Supporting information

**S1 Dataset.**
(XLSX)

**S1 Fig. HPAEC-PAD sialic acid.**
(XLSX)

## Acknowledgments

Mass spectrometry analysis was carried out at the Mass Spectrometry Core Facility of Faculty of Medicine and Health Sciences, Linköping University.

## Author Contributions

**Conceptualization:** Eva Landberg.

**Formal analysis:** Eva Landberg.

**Funding acquisition:** Eva Landberg.

**Investigation:** Daniel Hirschberg, Eva Landberg.

**Methodology:** Daniel Hirschberg.

**Resources:** Bertil Ekman, Jeanette Wahlberg.

**Validation:** Daniel Hirschberg.

**Writing – review & editing:** Daniel Hirschberg, Bertil Ekman, Eva Landberg.

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
