## [Decision Letter · Decision Letter 0]

24 Nov 2020

PONE-D-20-32839

Altered immunoglobulin G glycosylation in patients with isolated hyperprolactinaemia

PLOS ONE

Dear Dr. Landberg,

Thank you for submitting your manuscript to PLOS ONE. After careful consideration, we feel that it has merit but does not fully meet PLOS ONE’s publication criteria as it currently stands. Therefore, we invite you to submit a revised version of the manuscript that addresses the points raised during the review process.

The statistical analysis has caught the attention of both reviewers and the reported confusion should be cleared with a better justification of the approach and of the interpretation of results.

We look forward to receiving your revised manuscript.

Kind regards,

Frederique Lisacek

Academic Editor

PLOS ONE

Journal Requirements:

3. In your Methods section, please provide additional information about the participant recruitment method and the demographic details of your participants. Please ensure you have provided sufficient details to replicate the analyses such as: a) the recruitment date range (month and year), b) a table of relevant demographic details and c) a statement as to whether your sample can be considered representative of a larger population.

4. Please provide a sample size and power calculation in the Methods, or discuss the reasons for not performing one before study initiation.

Reviewers' comments:

Reviewer's Responses to Questions

**Comments to the Author**

1. Is the manuscript technically sound, and do the data support the conclusions?

Reviewer #1: Yes

Reviewer #2: Partly

2. Has the statistical analysis been performed appropriately and rigorously? 

Reviewer #1: Yes

Reviewer #2: No

3. Have the authors made all data underlying the findings in their manuscript fully available?

Reviewer #1: Yes

Reviewer #2: No

4. Is the manuscript presented in an intelligible fashion and written in standard English?

Reviewer #1: Yes

Reviewer #2: Yes

5. Review Comments to the Author

Reviewer #1: The authors investigated the IgG glycosylation changes in patients with hyperprolactinaemia. Moderate hyperprolactinaemia was found to be associated with increased galactosylation of IgG1, IgG2 and IgG3 though it wasn't confirmed in case of severe hyperprolactinaemia. Level of prolactin may have an impact on IgG interactions with Fc-receptors, complement and lectins, and consequently, lead to an altered immune response. The article is well written and structured, but we have some suggestions which could improve the quality of this manuscript.

Introduction: Macroprolactinaemia, which is mentioned in the text, should be also explained next to hyperprolactinaemia (line 73).

Lines 90-100: The description of different patient groups (including macroprolactinaemia group) is somewhat confusing at times, it would be better to include these data in the Table 1.

Line 192-193: Only fucosylated galactosylated glycans were taken into the equation to calculate galactosylation levels, but the IgG contains also non fucosylated species in lower amounts, as correctly commented on lines 240-241, were only major tryptic glycopeptides considered? This should be commented on.

Lines 273-274: What levels of prolactin were found in macroprolactin group? It would be useful to have these data for comparison, this information could be included in the Table 1.

Figures 2-3: Capital letters were used to mark panels on the figures (A-D), however, in the text with figure legends, small letters are used for these panels (a-d). Please, make it consistent in text and figures.

Line 120: The term of ‘immunobased’ is not correct please, use ’immune based’ .

Line 223-224: Patient group has significantly increased IgA, it should be mentioned that prolactin stimulates IgA production (1) and commented on.

Lines 294-295: Why the patients with macroprolactin (did you mean macroprolactinaemia patients?) were not included in correlation between prolactin and IgG2/3 galactosylation?

Line 363: Macroprolactin is a complex of immunoglobulin and prolactin. Was this macroprolactin captured together with other IgGs on the Protein G Sepharose column during immunoglobulin enrichment? If so, some of the analysed IgGs might be from this complex, this should be discussed. Macrolactin could be also an interesting object of follow up further investigation of glycomics changes and/or possible underlying glycan motif behind this complex formation associated with hyper/macroprolactinaenia.

Lines 356-362: IgG can also be O-glycosylated, so some sialic acid could come from there, this could be added to the discussion about sialylation.

Typos to be corrected: Line 243- older, line 341-menopause

Only IgG glycosylation was investigated, but the glycosylation on the other immunoglobulins maybe also affected in hyperprolactinaemia, this could be also included in the discussion.

References are not uniformed. Please, remove/or add all the DOI numbers after a referred publication.

Conclusion. After addressing these minor comments this manuscript will be suitable for publication in the PLOS ONE.

References.

1. Hermann G, O’Dorision MS. Modulation of IgA synthesis by neuroendocrine peptides. Trends Endocrinol Metab, 1991, 2(2): 68-72.

Reviewer #2: The authors examine the immunoglobulin G glycosylation pattern in patients with hyperprolactinaemia compared to healthy controls. The topic is interesting, unexplored and relevant, as it might provide insight into the relationship of this sex hormone with IgG glycome, more precisely, the proposed regulation of IgG glycosylation by estrogen. They examine in total 47 patients, divided into age- and sex-matched patient and control groups of similar subject number. They analyze the total IgG sialic acid content and subclass specific IgG Fc galactose and bisecting GlcNAc content. The results are more or less well presented, but I have a major issue with how the data were analyzed. Firstly, the patient group (25 subjects) appears way to small to be further divided into 3 subgroups. We are given no info on whether the subgroups are age- and sex- matched. Even if this were the case, this subdivision significantly increases the number of statistical tests performed. Which brings me to my second objection: no correction for multiple testing is performed. From the methods and results description, close to 30 statistical tests were performed and I am afraid some (if not all) p values deemed statistically significant might get lost after the correction. Unless there is a strong reason (is only a mild increase in prolactin level biologically relevant compared to very high values?), I would suggest only performing statistical analysis on the two main groups (patient vs. control) and re-assessing the results. Additionally, looking into the correlation of IgG glycosylation traits with prolactin levels on such a small sample number also seems problematic if age- and sex- correction is not previously performed on IgG glycan data. In summary, although the low sample number might not be sufficient to capture the small effect size that might be in effect here, I think this is an interesting pilot study and I believe it merits publication once the data is re-analyzed.

Minor issues:

- Some typos and inconsistencies in style.

- The common Fc glycopeptide of IgG2 and IgG3 subclass should be denoted as IgG2/3 at all times, to avoid ambiguity.

- Original glycan data (individual structures) not provided.

- Since the number of samples is small, I think the graphs would profit from single dot=subject representation in addition to characteristic descriptive values.

- Line 167: should it be “200 ug” instead of “200 mg”?

- Line 222: should it be “distributed” instead of “disturbed”?

- Line 338: I don’t think we can infer any causality from this study.

- A comment/question for the Discussion section: could correlation between prolactin levels and IgG glycoslyation traits be additionally “blurred” by prolactio concentration variation during menstrual cycle?

6. PLOS authors have the option to publish the peer review history of their article (what does this mean?). If published, this will include your full peer review and any attached files.

Reviewer #1: **Yes: **Dr Radka Fahey (Saldova) and Dr Zsuzsanna Kovacs

Reviewer #2: No

---

## [Author Response · Author response to Decision Letter 0]

28 Dec 2020

Answers to PLOS ONE editor and reviewers comments

PONE-D-20-32839

Altered immunoglobulin G glycosylation in patients with isolated hyperprolactinaemia

PLOS ONE

Please, find our answers to each comment below. The given line numbers refer to the file: “Revised Manuscript with track changes”. All changes are marked in red. 

The style requirements have been checked. The formulas in the Method section have been changed to Equation format, lines: 211 and 218.

Captions for supporting files have been added, lines: 649-651. 

3. In your Methods section, please provide additional information about the participant recruitment method and the demographic details of your participants. Please ensure you have provided sufficient details to replicate the analyses such as: a) the recruitment date range (month and year), b) a table of relevant demographic details and c) a statement as to whether your sample can be considered representative of a larger population.

Additional information about the recruitment of the patients and controls has been added to the Method section, lines: 89-93 and 113-114.

Relevant demographic details are presented in Table 1 and for the patients with macroprolactinaemia in the Result section, lines: 240-241.

4. Please provide a sample size and power calculation in the Methods, or discuss the reasons for not performing one before study initiation.

Sample size and power calculation has been added to the Material and Method section, lines: 206-208.

Reviewers' comments:

Reviewer's Responses to Questions

Comments to the Author

1. Is the manuscript technically sound, and do the data support the conclusions?

Reviewer #1: Yes

Reviewer #2: Partly

2. Has the statistical analysis been performed appropriately and rigorously? 

Reviewer #1: Yes

Reviewer #2: No

3. Have the authors made all data underlying the findings in their manuscript fully available?

Reviewer #1: Yes

Reviewer #2: No

4. Is the manuscript presented in an intelligible fashion and written in standard English?

Reviewer #1: Yes

Reviewer #2: Yes

5. Review Comments to the Author

Reviewer #1: The authors investigated the IgG glycosylation changes in patients with hyperprolactinaemia. Moderate hyperprolactinaemia was found to be associated with increased galactosylation of IgG1, IgG2 and IgG3 though it wasn't confirmed in case of severe hyperprolactinaemia. Level of prolactin may have an impact on IgG interactions with Fc-receptors, complement and lectins, and consequently, lead to an altered immune response. The article is well written and structured, but we have some suggestions which could improve the quality of this manuscript.

Thanks for your suggestions, which certainly will improve the manuscript. Answers to each comment are included in red below. 

Introduction: Macroprolactinaemia, which is mentioned in the text, should be also explained next to hyperprolactinaemia (line 73).

This information has been added to the Introduction section, lines: 76-78.

Lines 90-100: The description of different patient groups (including macroprolactinaemia group) is somewhat confusing at times, it would be better to include these data in the Table 1.

As Table 1 describes a comparison between hyperprolactinaemia patients (without macroprolactin) and controls, this information is difficult to include in the table as suggested. Instead, we have reformulated the text in the Method section, which hopefully will make the description more clearly, lines: 105-112.

Line 192-193: Only fucosylated galactosylated glycans were taken into the equation to calculate galactosylation levels, but the IgG contains also non fucosylated species in lower amounts, as correctly commented on lines 240-241, were only major tryptic glycopeptides considered? This should be commented on.

Comment on this has been added to the Method section, lines: 214-215.

Lines 273-274: What levels of prolactin were found in macroprolactin group? It would be useful to have these data for comparison, this information could be included in the Table 1.

This information has been included in the text as follows, lines: 237-241.

Figures 2-3: Capital letters were used to mark panels on the figures (A-D), however, in the text with figure legends, small letters are used for these panels (a-d). Please, make it consistent in text and figures.

Small letters are now used consistently. 

Line 120: The term of ‘immunobased’ is not correct please, use ’immune based’ .

Corrected, line: 135.

Line 223-224: Patient group has significantly increased IgA, it should be mentioned that prolactin stimulates IgA production (1) and commented on.

Comment on this and the proposed reference (Hermann G et al., 1991) has been added to the Discussion section, lines 400-402.

Lines 294-295: Why the patients with macroprolactin (did you mean macroprolactinaemia patients?) were not included in correlation between prolactin and IgG2/3 galactosylation?

Corrected caption Fig 4 (now Fig 3), lines: 331-332.

An explanation has also been added to the Result section, lines: 340-342.

Line 363: Macroprolactin is a complex of immunoglobulin and prolactin. Was this macroprolactin captured together with other IgGs on the Protein G Sepharose column during immunoglobulin enrichment? If so, some of the analysed IgGs might be from this complex, this should be discussed. Macrolactin could be also an interesting object of follow up further investigation of glycomics changes and/or possible underlying glycan motif behind this complex formation associated with hyper/macroprolactinaenia.

Comments on this has been added to the Discussion section, lines: 432-435.

Lines 356-362: IgG can also be O-glycosylated, so some sialic acid could come from there, this could be added to the discussion about sialylation.

Comment on this and reference (29; Plomp R et al, 2015) has been added to the Discussion section, lines: 416-417.

Typos to be corrected: Line 243- older, line 341-menopause

Corrected, lines: 395 and 393. 

Only IgG glycosylation was investigated, but the glycosylation on the other immunoglobulins maybe also affected in hyperprolactinaemia, this could be also included in the discussion.

Comments on this has been added to the Discussion section, lines: 419-422.

References are not uniformed. Please, remove/or add all the DOI numbers after a referred publication.

DOI numbers (if existing) have been added to the References. 

Conclusion. After addressing these minor comments this manuscript will be suitable for publication in the PLOS ONE.

References.

1. Hermann G, O’Dorision MS. Modulation of IgA synthesis by neuroendocrine peptides. Trends Endocrinol Metab, 1991, 2(2): 68-72.

Reviewer #2: The authors examine the immunoglobulin G glycosylation pattern in patients with hyperprolactinaemia compared to healthy controls. The topic is interesting, unexplored and relevant, as it might provide insight into the relationship of this sex hormone with IgG glycome, more precisely, the proposed regulation of IgG glycosylation by estrogen. They examine in total 47 patients, divided into age- and sex-matched patient and control groups of similar subject number. They analyze the total IgG sialic acid content and subclass specific IgG Fc galactose and bisecting GlcNAc content. The results are more or less well presented, but I have a major issue with how the data were analyzed. Firstly, the patient group (25 subjects) appears way to small to be further divided into 3 subgroups. We are given no info on whether the subgroups are age- and sex- matched. Even if this were the case, this subdivision significantly increases the number of statistical tests performed. Which brings me to my second objection: no correction for multiple testing is performed. From the methods and results description, close to 30 statistical tests were performed and I am afraid some (if not all) p values deemed statistically significant might get lost after the correction. Unless there is a strong reason (is only a mild increase in prolactin level biologically relevant compared to very high values?), I would suggest only performing statistical analysis on the two main groups (patient vs. control) and re-assessing the results. Additionally, looking into the correlation of IgG glycosylation traits with prolactin levels on such a small sample number also seems problematic if age- and sex- correction is not previously performed on IgG glycan data. In summary, although the low sample number might not be sufficient to capture the small effect size that might be in effect here, I think this is an interesting pilot study and I believe it merits publication once the data is re-analyzed.

Thanks for your suggestions, which certainly will improve the manuscript. Answers to each comment are included in red below. 

This study was designed to examine IgG-N-glycosylation in patients with hyperprolactinaemia compared to normoprolactinaemic healthy controls. The analyses presented in Table 1 were performed to verify similarity between groups minimizing possible confounding factors. Statistics in this part of the study was therefore designed to minimize type II errors (false negatives) and not to minimize type I errors (risk of incorrectly reject a true 0-hypothesis). Correction for multiple testing is thus not applicable. However, significant differences in this table should be interpreted with caution. If using Bonferroni correction for multiple testing both IgA and prolactin still show significant differences between groups. Age and gender were not considered as parameters in the Bonferroni correction as these are matched entities. 

A corrected p-value for IgA has been added to the Result section, line 251.

To further explain the statistic approach, following sentences have been added to the Statistical section, line 226-231.

Considering the four IgG-glycosylation parameters, there is a significant correlation between IgG1 galactosylation and IgG2/3 galactosylation within groups. The sialic acid content on IgG is also likely correlated to IgG galctosylation, though we did not find a significant correlation in this study. To our knowledge, correction for multiple testing should not be performed in these instances. 

We agree that by dividing the patient group in two (three including the macroprolactin group) will result in few individuals in each group and a weak statistical power. We have therefore excluded these results and Fig 3. The results from the macroprolactinaemic group are included in Fig 2 for visual comparison but without performing statistical measurements. It is emphasized in the discussion that glycosylation differences in this group of patients needs confirmation using a larger group of patients, lines: 429-430.

Accordingly, several parts in the Abstract, Result and Discussion section have been changed. Lines: 34-36, 39-41, 278-280, 300-324, 335-342, 385-387, 428-429, 445-448. 

Minor issues:

- Some typos and inconsistencies in style.

- The common Fc glycopeptide of IgG2 and IgG3 subclass should be denoted as IgG2/3 at all times, to avoid ambiguity.

Corrected to IgG2/3. 

- Original glycan data (individual structures) not provided.

Data from MALDI-MS spectra (intensity per m/z value) have been added (S1 Dataset)

A HPAEC-PAD chromatogram from analysis of sialic acid from one sample (including a standard) has been added, line 277, S2 Figure. 

- Since the number of samples is small, I think the graphs would profit from single dot=subject representation in addition to characteristic descriptive values.

Fig 2 has been changed, now also presenting single dot=subject representation. 

- Line 167: should it be “200 ug” instead of “200 mg”?

Corrected, line: 182. 

- Line 222: should it be “distributed” instead of “disturbed”?

Corrected, line: 249.

- Line 338: I don’t think we can infer any causality from this study.

The word “strengthens” has been replaced by “indicates” in this sentence, line: 383.

- A comment/question for the Discussion section: could correlation between prolactin levels and IgG glycoslyation traits be additionally “blurred” by prolactio concentration variation during menstrual cycle?

A paragraph describing biological variation of prolactin has been added to the discussion, lines: 387-392

Two references has also been added (24; Freeman ME et al., 2000 and 26; Fujimoto VU et al., 1990). 

---

## [Decision Letter · Decision Letter 1]

15 Feb 2021

Altered immunoglobulin G glycosylation in patients with isolated hyperprolactinaemia

PONE-D-20-32839R1

Dear Dr. Landberg,

We’re pleased to inform you that your manuscript has been judged scientifically suitable for publication and will be formally accepted for publication once it meets all outstanding technical requirements.

Kind regards,

Frederique Lisacek

Academic Editor

PLOS ONE

Additional Editor Comments (optional):

Reviewers' comments:

Reviewer's Responses to Questions

**Comments to the Author**

1. If the authors have adequately addressed your comments raised in a previous round of review and you feel that this manuscript is now acceptable for publication, you may indicate that here to bypass the “Comments to the Author” section, enter your conflict of interest statement in the “Confidential to Editor” section, and submit your "Accept" recommendation.

Reviewer #1: All comments have been addressed

Reviewer #2: All comments have been addressed

2. Is the manuscript technically sound, and do the data support the conclusions?

Reviewer #1: Yes

Reviewer #2: Yes

3. Has the statistical analysis been performed appropriately and rigorously? 

Reviewer #1: Yes

Reviewer #2: Yes

4. Have the authors made all data underlying the findings in their manuscript fully available?

Reviewer #1: Yes

Reviewer #2: Yes

5. Is the manuscript presented in an intelligible fashion and written in standard English?

Reviewer #1: Yes

Reviewer #2: Yes

6. Review Comments to the Author

Reviewer #1: The authors implemented all suggested modifications. The authors' work had improved the quality of the manuscript and we support the acceptance of the manuscript for publication in the journal of PLOS ONE.

We have only one minor suggestion- the numbering of equations on lines 211 and 218 in the revised manuscript with tracked changes are rather confusing as they do not seem to be referenced in the text, are these labels necessary?

Reviewer #2: The authors have addressed all of my comments and I believe the manuscript is now ready for publication.

7. PLOS authors have the option to publish the peer review history of their article (what does this mean?). If published, this will include your full peer review and any attached files.

Reviewer #1: No

Reviewer #2: No

---

## [Editor Report · Acceptance letter]

18 Feb 2021

PONE-D-20-32839R1 

Altered immunoglobulin G glycosylation in patients with isolated hyperprolactinaemia 

Dear Dr. Landberg:

I'm pleased to inform you that your manuscript has been deemed suitable for publication in PLOS ONE. Congratulations! Your manuscript is now with our production department. 

Kind regards, 

on behalf of

Dr. Frederique Lisacek 

Academic Editor

PLOS ONE